# MULTI-LABEL CLUSTER DISCRIMINATION FOR VISUAL REPRESENTATION LEARNING

## ABSTRACT

Contrastive Language-Image Pre-training (CLIP) has recently demonstrated success across various tasks due to superior feature representation empowered by image-text contrastive learning. However, the instance discrimination method used by CLIP can hardly encode the semantic structure of training data. To handle this limitation, cluster discrimination has been proposed through iterative classification and cluster assignment. Nevertheless, most cluster discrimination approaches only define a single pseudo-label for each image, neglecting multi-label signals in the image. In this paper, we propose a novel multi-label cluster discrimination method to enhance representation learning. In the clustering step, we first cluster the large-scale LAION 400M dataset into one million centers based on off-the-shelf embedding features. Considering that natural images frequently contain multiple visual targets, we select the multiple closest centers as additional class labels. In the discrimination step, we design an efficient multi-label classification loss, which elegantly separates losses from positive classes and negative classes and facilitates distributed training on large-scale data. We validate the proposed multi-label cluster discrimination method with experiments on different scales of models and pre-training datasets. Experimental results show that our method achieves state-of-the-art performance on multiple downstream tasks including linear probe, zero-shot classification, and image-text retrieval.

## 1 INTRODUCTION

Language-supervised visual pre-training, e.g., CLIP (Radford et al., 2021) and ALIGN (Jia et al., 2021), has been established as a simple yet effective methodology for visual representation learning. Empowered by image-text contrastive learning, pre-trained CLIP models exhibit remarkable versatility and transferability across various downstream tasks (e.g., linear probe, zero-shot classification, and image retrieval). As illustrated in Fig. 1a, CLIP aligns the visual and textual signals of each instance into a unified semantic space by cross-modal instance discrimination. Nevertheless, the instance discrimination method used by CLIP can hardly encode the semantic structure of training data, because instance-wise contrastive learning always treats two samples as a negative pair if they are from different instances, regardless of their semantic similarity. When a large number of instances are selected into the mini-batch to form the contrastive loss, negative pairs that share similar semantics will be undesirably pushed apart in the embedding space.

To handle the limitations of instance discrimination, cluster discrimination methods (e.g., DeepCluster (Caron et al., 2018), SeLa (Asano et al., 2020), ODC (Zhan et al., 2020), SwAV (Caron et al., 2020), CoKe (Qian et al., 2022), and UNICOM (An et al., 2023)) have been proposed for deep unsupervised learning through jointly learning image embeddings and cluster assignments. Learning representations with clusters will pull similar instances together, which is beneficial for capturing semantic structures in data. However, most cluster discrimination approaches only define a single pseudo-label for each image as depicted in Fig. 1b. By contrast, natural language supervision proposed in CLIP can provide richer forms of labels for a single image, e.g., objects, scenes, actions, and relations, at multiple levels of granularity.

As can be seen from Fig. 2, a web image frequently contains multiple classification targets, such as objects (Yang et al., 2016) and attributes (Pham et al., 2021). The existence of multiple objects in the image requires laborious cropping (Li et al., 2023a; Abdelfattah et al., 2023) to construct single-label

annotations, while some scenario elements in the image are hard to disentangle to obtain single-label instances (Pham et al., 2021; Zhu et al., 2023). These real-world challenges pose so-called multi-label classification where an image is equipped with multiple labels beyond a single label.

In this paper, we aim at boosting the visual representation power of the CLIP model by introducing a novel Multi-Label Cluster Discrimination (MLCD) approach. In the **clustering step**, we follow UNICOM (An et al., 2023) to conduct one step of off-line clustering by using the features predicted by a pre-trained CLIP model (Radford et al., 2021). Due to the limited discrimination power of the CLIP model, the single pseudo-label may not cover all of the visual signals in the image. To this end, we further perform a similarity-based sorting against $k$ class centers and select the top $l$ class centers as the positive class centers for that image. In the **discrimination step**, we follow the Circle loss (Sun et al., 2020) to design a multi-label loss to effectively deal with multiple labels. The vanilla version of the multi-label loss exploits relative similarity comparison between positive and negative classes. More specifically, the optimization seeks to narrow the gap between the intra-class similarities $\{s_i\}$ and the inter-class similarities $\{s_j\}$ by reducing all possible $(s_j - s_i)$. However, optimizing $(s_j - s_i)$ usually leads to a decision boundary allowing ambiguity (Sun et al., 2020). To this end, we introduce another two optimization targets (i.e., decreasing $s_j$ and increasing $s_i$) into the loss function. Introducing the additional two items enables an elegant separation of positive class loss and negative class loss (Eq. 5), which can facilitate distributed training on large-scale data with minimal communication overhead. To alleviate inter-class conflict and save the computation time on the classifier layer, we also employ PartialFC (An et al., 2022) and randomly sample part of the negative class centers during each iteration.

The main contributions of our paper are the following:

- We propose a novel multi-label cluster discrimination method for visual representation learning on large-scale data. In the clustering step, we employ one step of offline k-means to predict multiple labels for each training sample. In the discrimination step, we explore multi-label classification, which considers multiple supervision signals for a single image and learns better semantic structure in data.
- To avoid ambiguity during the optimization of $(s_j - s_i)$, we add additional optimization targets by maximizing the within-class similarity $s_i$, as well as to minimizing the between-class similarity $s_j$. By doing so, the loss from positive class labels and negative class labels can be elegantly separated, decreasing the communication cost of distributed training.
- The proposed multi-label cluster discrimination significantly boosts the representation power compared to the instance discrimination-based model (e.g., OpenCLIP and FLIP (Li et al., 2023b)) and the cluster discrimination-based model (e.g., UNICOM (An et al., 2023)) on the downstream tasks (e.g., linear probe, zero-shot classification, zero-shot retrieval).

## 2 RELATED WORK

**Visual Representation Learning.** Visual representation pre-training methods can be mainly divided into three categories: (1) supervised learning by using manually annotated class labels (e.g., ImageNet-1K/-21K (Deng et al., 2009) and JFT-300M/-3B (Dosovitskiy et al., 2021; Zhai et al., 2022a)), (2) weakly-supervised learning by employing hashtags (Mahajan et al., 2018; Singh et al., 2022) or text descriptions (Radford et al., 2021; Jia et al., 2021; Li et al., 2023b), and (3) unsupervised learning (Chen et al., 2020; He et al., 2020; Caron et al., 2018) by designing appropriate pretext tasks (e.g., solving jigsaw puzzles (Noroozi & Favaro, 2016), invariant mapping (Chen & He, 2021), and masked image inpainting (He et al., 2022)). Even though fully supervised pre-training can learn a strong semantic signal from each training example, manual label annotation is time-consuming and expensive thus supervised learning is less scalable. In this paper, we focus on annotation-free pre-training which can be easily scaled to billions of web images to learn visual representation for downstream tasks.

**Instance and Cluster Discrimination.** Instance discrimination (Chen et al., 2020; He et al., 2020; Radford et al., 2021) is usually implemented by the contrastive loss to pull images from the same instance as well as push away images from different instances. Among these instance discrimination methods, language-supervised visual pre-training, e.g., CLIP (Radford et al., 2021), is a simple yet powerful approach to take advantage of rich forms of labels at multiple levels of granularity for a single image. Even though CLIP (Radford et al., 2021) has recently demonstrated impressive

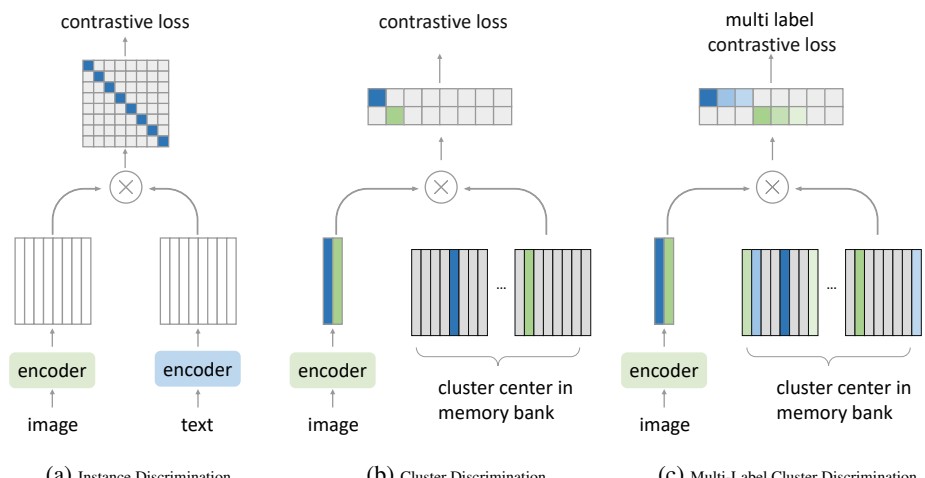

(a) Instance Discrimination    (b) Cluster Discrimination    (c) Multi-Label Cluster Discrimination

Figure 1: Overview of our multi-label cluster discrimination method. Fig. 1a treats each image-text pair as a unique instance, failing to capture the data's semantic structure. Fig. 1b improves by grouping similar instances but struggles with multi-label signals in a single image. Fig. 1c, our proposed method, addresses this challenge by assigning multiple class labels to each sample, capturing different granularities of visual signals in one image.

success, instance-wise contrastive learning always treats different instances as negative pairs thus it can hardly capture the full semantic information from the training data. To explore potential semantic structures in the training data, cluster discrimination (Caron et al., 2018; Asano et al., 2020; Zhan et al., 2020; Li et al., 2020; Caron et al., 2020; Qian et al., 2022) is proposed with two iterative steps: (1) the clustering step to assign a single class label for each sample, and (2) the classification step to learn a classifier to predict the assigned pseudo label. In cluster discrimination methods, each cluster contains more than one instance, visually similar instances will be pulled closer and thus cluster discrimination can better capture semantic structures from data. However, multiple visual elements can exist in one single image and the single label used by cluster discrimination may not cover all visual signals.

**Multi-label Classification.** Multi-label classification (Tsoumakas & Katakis, 2007; Zhang & Zhou, 2013) assigns a set of multiple labels for each instance. Compared with single-class classification, where each instance is assigned with a single label, multi-label classification (Yang et al., 2016; Zhao et al., 2021; Xia et al., 2023) is more challenging (Liu et al., 2017; 2021). Considering multiple labels are drawn from $k$ categories, the multi-label classification can be decomposed into $k$ binary classification tasks. However, the binary cross-entropy loss involves issues regarding imbalance (Ridnik et al., 2021). Through analyzing the intrinsic loss functions of the classification loss and the metric loss (Wang et al., 2019), Sun et al. (Sun et al., 2020) formulate a unified multi-label loss function to exploit relative comparison between positive and negative classes. Nevertheless, the relative comparison $(s_j - s_i)$ allows ambiguity for convergence. In this paper, we only employ one step of offline clustering to predict multiple labels for each image and then design an efficient and robust multi-label classifier to achieve good feature representation when training on the automatically clustered large-scale data.

## 3 METHODOLOGY

Given a training set $X = \{x_1, x_2, ..., x_n\}$ including $n$ images, visual representation learning aims at learning a function $f$ that maps images $X$ to normalized embeddings $E = \{e_1, e_2, ..., e_n\}$ with $e_i = f(x_i)$, such that embeddings can describe the semantic similarities between different images.

### 3.1 PRELIMINARIES

**Instance Discrimination** achieves semantic embedding by minimizing a contrastive loss function represented as:

$$\mathcal{L}_{\text{ID}} = -\log \frac{\exp(e_i'^T e_i)}{\sum_{j=1}^{k} \exp(e_j'^T e_i)}, \tag{1}$$

where $\exp(\cdot)$ denotes the exponential function, and $e_i$ and $e_i'$ denote the normalized image and text embeddings for the instance $i$ in CLIP (Radford et al., 2021). Meanwhile, $e_j'$ contains one positive text representation for $i$ and $(k-1)$ negative text representations sourced from different instances. As illustrated in Fig. 1a, the instance discrimination based CLIP model jointly trains an image encoder and a text encoder to predict the correct image-text pairings from a batch of training examples.

**Cluster Discrimination** is composed of two primary stages: the clustering process and the discrimination process. During the clustering phase, every instance is assigned one pseudo-class label. This label is later employed as a guiding factor for training a classifier in the subsequent discrimination phase. For the normalized embedding feature $e_i = f(x_i)$, the clustering process determines a centroid matrix $W \in \mathbb{R}^{d \times k}$ and assigns the cluster label $y_i$ for each image $x_i$. This is achieved by

$$\min_{W \in \mathbb{R}^{d \times k}} \frac{1}{n} \sum_{i=1}^{n} \min_{y_i \in \{0,1\}^k} \|e_i - W y_i\|_2^2 \quad \text{s.t.} \quad y_i^\top \mathbf{1_k} = \mathbf{1}, \tag{2}$$

where $n$ is the number of training samples, $e_i$ is the normalized feature embedding obtained by using the image encoder $f$, and the centroid $w_i$ belonging to centroid matrix $W \in \mathbb{R}^{d \times k}$ is considered the normalized prototype of $i$-th cluster. $y_i$, falling within the set $\{0,1\}^k$, stands as a single label assignment restricted by the condition $y_i^\top \mathbf{1_k} = \mathbf{1}$, where $\mathbf{1_k}$ is 1-vector with a length of $k$.

Then, the training data, denoted as $\{x_i\}_{i=1}^n$, is divided into $k$ classes represented by prototypes $W = \{w_i\}_{i=1}^k$. Utilizing the pseudo labels and centroids derived from the clustering phase, the process of cluster discrimination can be executed by minimizing a conventional softmax classification loss, formulated as:

$$\mathcal{L}_{\text{CD}} = -\log \frac{\exp(w_i^T e_i)}{\sum_{j=1}^{k} \exp(w_j^T e_i)} = -\log \frac{\exp(s_i)}{\sum_{j=1}^{k} \exp(s_j)} = \log(1 + \sum_{j=1, j \neq i}^{k} \exp(s_j - s_i)), \tag{3}$$

where $e_i$ is the normalized embedding corresponding to the image $x_i$, and $x_i$ is categorized under the class symbolized by the normalized prototype $w_i$. For a more straightforward representation, we define the intra-class similarity $w_i^T e_i$, and the inter-class similarity, $w_j^T e_i$ as $s_i$ and $s_j$, respectively. Based on Eq. 3, in the discrimination phase that employs classification, $s_j$ and $s_i$ are paired to optimize the reduction of the difference $(s_j - s_i)$. As depicted in Fig. 1b, the cluster discrimination based UNICOM model (An et al., 2023) trains an image encoder to predict the one-hot pseudo label for each image from a batch of training examples.

## 3.2 MULTI-LABEL CLUSTER DISCRIMINATION

**Clustering.** Considering the time consumption of iterative clustering and discrimination (Caron et al., 2018), An et al. (An et al., 2023) implemented a single step of offline clustering with the aid of efficient feature quantization (Johnson et al., 2019). On the large-scale LAION-400M dataset, it only takes around 10 minutes to cluster one million classes. Despite the straightforwardness of the clustering step, the automatically clustered large-scale dataset inevitably confronts intra-class purity and inter-class conflict problems due to the specific definition of class granularity.

In the realm of clustering algorithms, there often exists a trade-off between maintaining high within-class purity and ensuring low inter-class conflict. In the context of contrastive learning, the issue of inter-class conflict can be significantly alleviated by reducing the number of sampled negative instances within the mini-batch and adopting a suitable semi-hard mining technique. In this paper, we follow UNICOM (An et al., 2023) to prioritize intra-class purity (i.e., clustering one million level classes from 400 million images) and employ PatialFC (An et al., 2022) to alleviate inter-class conflict (i.e., randomly sampling part of the negative class centers during each iteration).

**Multi-label Classification.** As illustrated in Fig. 2, a single image can encompass several visual components. This implies that even if one class only contains one image, the single class label may not cover all visual cues present in the image. To consider the different granularities of visual information, for each sample, we perform a similarity-based sorting against one million class centers, selecting the top $l$ class centers as the positive class centers for that sample. During training, this

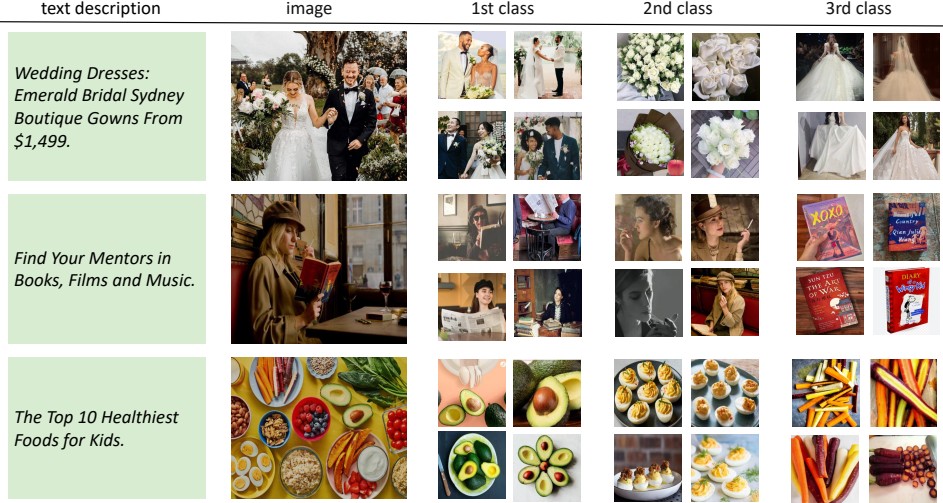

Figure 2: Illustration of the multiple visual elements in images from the automatically clustered LAION-400M dataset.

sample will be directed to move closer to these $l$ positive class centers, while simultaneously distancing from the other $k-l$ negative class centers. As shown in Fig. 1c, our method assigns multiple class labels to each training example, capturing different granularities of visual signals in one image.

The corresponding similarity scores are represented as $\{s_i\}$ $(i = 1, 2, \cdots, l)$ and $\{s_j\}$ $(j = 1, 2, \cdots, k - l)$, respectively. To minimize each $s_j$ as well as to maximize $s_i$, $(\forall i \in \{1, 2, \cdots, l\}, \forall j \in \{1, 2, \cdots, k - l\})$, we employ a multi-label classification strategy (Li et al., 2017; Sun et al., 2020). This is achieved by

$$\mathcal{L}_{\text{MLC}} = \log(1 + \underbrace{\sum_{j=1}^{k-l}\sum_{i=1}^{l}\exp(s_j - s_i)}_{contrastive}) = \log(1 + \underbrace{\sum_{j\in\Omega_n}\exp(s_j)\sum_{i\in\Omega_p}\exp(-s_i)}_{contrastive}), \quad (4)$$

where $\Omega_n$ and $\Omega_p$ denote the negative and positive class set in order to simplify the representation. Here, Eq. 4 iterates through every similarity pair to reduce $(s_j - s_i)$. Optimizing $(s_j - s_i)$ usually leads to a decision boundary of $s_j - s_i = m$ ($m$ is the margin). However, this decision boundary allows ambiguity as indicated in Circle loss (Sun et al., 2020). For example, $\{s_j, s_i\} = \{0.1, 0.4\}$ and $\{s'_j, s'_i\} = \{0.5, 0.8\}$ both achieve the margin $m = 0.3$. However, the gap between $s_i$ and $s'_j$ is only 0.1, compromising the separability of the feature space. As we expect to maximize the within-class similarity $s_i$ and to minimize the between-class similarity $s_j$, we further introduce these two items into the multi-label classification loss:

$$\mathcal{L}_{\text{MLCD}} = \log(1 + \underbrace{\sum_{j\in\Omega_n}\exp(s_j)\sum_{i\in\Omega_p}\exp(-s_i)}_{contrastive} + \underbrace{\sum_{j\in\Omega_n}\exp(s_j)}_{negative} + \underbrace{\sum_{i\in\Omega_p}\exp(-s_i)}_{positive})$$

$$= \log(1 + \sum_{i\in\Omega_p}\exp(-s_i)) + \log(1 + \sum_{j\in\Omega_n}\exp(s_j)), \quad (5)$$

where $\Omega_p$ symbolizes the collection of positive class labels for each sample, $s_i$ encapsulates the score associated with each positive class, $\Omega_n$ denotes the collection of negative class labels for each sample, and $s_j$ corresponds to the score for each negative class. In Eq. 5, loss from positive class labels $\log(1 + \sum_{i\in\Omega_p}\exp(-s_i))$ and loss from negative class labels $\log(1 + \sum_{j\in\Omega_n}\exp(s_j))$ are elegantly separated. To alleviate inter-class conflict as in (An et al., 2022; 2023), we also employ negative class sampling into Eq. 5,

$$\mathcal{L}'_{\text{MLCD}} = \log(1 + \sum_{i\in\Omega_p}\exp(-s_i)) + \log(1 + \sum_{j\in\Omega'_n}\exp(s_j)), \quad (6)$$

where $|\Omega'_n| = |\Omega_n| * r$, and $r \in [0, 1]$ is the negative class sampling ratio. $\Omega'_n$ is a subset of $\Omega_n$ that is randomly sampled during each loss calculation step.

| CASE | DATA | Food101 | CIFAR10 | CIFAR100 | Birdsnap | SUN397 | Cars | Aircraft | VOC2007 | DTD | Pets | Cal101 | Flowers | MNIST | FER2013 | STL10 | EuroSAT | RESISC45 | GTSRB | KITTI | Country211 | PCAM | UCF101 | K700 | CLEVR | HM | SST | AVG |
|---|---|---|---|---|---|---|---|---|---|---|---|---|---|---|---|---|---|---|---|---|---|---|---|---|---|---|---|---|
| CLIP† | WIT-400M | 95.2 | 98.0 | 87.5 | 77.0 | 81.8 | 90.9 | 69.4 | 89.6 | 82.1 | 95.1 | 96.5 | 99.2 | 99.2 | 72.2 | 99.8 | 98.2 | 94.1 | 92.5 | 64.7 | 42.9 | 85.8 | 91.5 | 72.0 | 57.8 | 76.2 | 80.8 | 84.2 |
| CLIP‡ | WIT-400M | 95.3 | 98.1 | 87.2 | 77.8 | 81.5 | 90.7 | 68.0 | 89.7 | 80.9 | 94.9 | 96.0 | 99.2 | 99.2 | 72.3 | 99.8 | 96.7 | 94.5 | 92.9 | 65.9 | 41.9 | 85.3 | 91.0 | 70.6 | 59.6 | 61.8 | 79.8 | 83.5 |
| OPNCLIP‡ | LAION-400M | 93.3 | 97.9 | 87.9 | 78.0 | 81.0 | 93.6 | 64.4 | 91.7 | 83.0 | 93.3 | 95.5 | 98.8 | 99.2 | 66.5 | 99.2 | 97.1 | 92.4 | 92.5 | 77.5 | 32.5 | 84.3 | 88.1 | 64.0 | 59.8 | 57.6 | 71.9 | 82.3 |
| UNICOM | LAION-400M | 93.4 | 98.5 | 90.8 | 82.4 | 80.0 | 94.6 | 74.5 | 91.4 | 82.2 | 94.2 | 95.7 | 99.3 | 99.2 | 68.7 | 98.5 | 96.7 | 92.6 | 92.7 | 77.8 | 33.4 | 85.4 | 87.4 | 66.7 | 60.3 | 57.4 | 72.4 | 83.3 |
| Ours | LAION-400M | 94.3 | 98.9 | 92.0 | 83.4 | 82.1 | 94.8 | 79.6 | 92.5 | 84.6 | 95.3 | 97.2 | 99.3 | 99.3 | 72.4 | 99.3 | 99.1 | 94.7 | 92.5 | 78.2 | 34.5 | 86.0 | 90.0 | 68.5 | 60.1 | 57.9 | 73.4 | 84.6 |

Table 1: Linear probe performance of various pre-trained models on 26 datasets. †: Results reported in CLIP paper. ‡: Results we reproduced. Entries in green are the best results using LAION-400M.

| CASE | DATA | Food101 | CIFAR10 | CIFAR100 | Birdsnap | SUN397 | Cars | Aircraft | VOC2007 | DTD | Pets | Cal101 | Flowers | MNIST | STL10 | EuroSAT | RESISC45 | GTSRB | KITTI | Country211 | PCAM | UCF101 | K700 | CLEVR | HM | SST | AVG |
|---|---|---|---|---|---|---|---|---|---|---|---|---|---|---|---|---|---|---|---|---|---|---|---|---|---|---|---|
| CLIP† | WIT-400M | 92.9 | 96.2 | 77.9 | 48.3 | 67.7 | 77.3 | 36.1 | 84.1 | 55.3 | 93.5 | 92.6 | 78.7 | 87.2 | 99.3 | 59.9 | 71.6 | 50.3 | 23.1 | 32.7 | 58.8 | 76.2 | 60.3 | 24.3 | 63.3 | 64.0 | 66.9 |
| CLIP‡ | WIT-400M | 91.0 | 95.2 | 75.6 | 51.2 | 66.6 | 75.0 | 32.3 | 83.3 | 55.0 | 93.6 | 92.4 | 77.7 | 76.0 | 99.3 | 62.0 | 71.6 | 51.6 | 26.9 | 30.9 | 51.6 | 76.1 | 59.5 | 22.2 | 55.3 | 67.3 | 65.6 |
| OpenCLIP‡ | LAION-400M | 87.4 | 94.1 | 77.1 | 61.3 | 70.7 | 86.2 | 21.8 | 83.5 | 54.9 | 90.8 | 94.0 | 72.1 | 71.5 | 98.2 | 53.3 | 67.7 | 47.3 | 29.3 | 21.6 | 51.1 | 71.3 | 50.5 | 22.0 | 55.3 | 57.1 | 63.6 |
| FLIP‡ | LAION-400M | 89.3 | 97.2 | 84.1 | 63.0 | 73.1 | 90.7 | 29.1 | 83.1 | 60.4 | 92.6 | 93.8 | 75.0 | 80.3 | 98.5 | 53.5 | 70.8 | 41.4 | 34.8 | 23.1 | 50.3 | 74.1 | 55.8 | 22.7 | 54.0 | 58.5 | 66.0 |
| Ours | LAION-400M | 90.3 | 95.3 | 83.7 | 62.9 | 72.1 | 90.1 | 39.4 | 84.5 | 62.3 | 93.7 | 93.9 | 79.4 | 78.5 | 99.1 | 59.7 | 69.9 | 50.7 | 28.7 | 27.9 | 53.7 | 75.7 | 57.7 | 22.2 | 58.4 | 57.9 | 67.5 |

Table 2: Zero-shot classification performance on 25 datasets. †: Results reported in CLIP paper. ‡: Results reported in FLIP paper. Entries in green are the best results using LAION-400M.

## 4 EXPERIMENTS

### 4.1 EXPERIMENTAL SETTING

Our models are pre-trained on the LAION-400M dataset (Schuhmann et al., 2021) with the same model configurations as CLIP. The training process consists of 32 epochs, utilizing a batch size of 32K on 80 NVIDIA A100 GPUs. To expedite the training, we employ mixed-precision computation (Micikevicius et al., 2017) and flash attention (Dao et al., 2022), while leveraging the DALI library for efficient data loading and preprocessing. We use the AdamW optimizer with a learning rate of 0.001 and weight decay of 0.2. To assess the performance of zero-shot classification and zero-shot image-text retrieval tasks, we employ contrastive learning to train a text encoder from scratch for 32 epochs with a frozen image encoder following Locked-image Tuning (LiT) (Zhai et al., 2022b). The structure of the text encoder is also identical to CLIP. In the following experiments, unless otherwise specified, the model used is ViT-L/14, the number of classes ($k$) is one million, the ratio of sampled negative class centers ($r$) is 0.1, and the number of positive labels ($l$) assigned to each image is 8.

### 4.2 LINEAR PROBE

Following the same evaluation setting as CLIP, we report the linear probe performance of our method on 26 datasets. As depicted in Tab. 1, inherent biases exist in different pre-training data. The WIT dataset is beneficial for action-related datasets (e.g., Kinetics700, UCF101), while LAION exhibits superior proficiency in object datasets (e.g., Cars, Birdsnap). Nevertheless, our method still achieves an average improvement of 1.1% compared to CLIP. To isolate the confounding effects of pre-training data, we compare our model with OPENCLIP and UNICOM by using the LAION-400M dataset as the training data. As shown in Fig. 3a, our method outperforms OPENCLIP on 25 datasets, demonstrating an average improvement of 2.3%. In Fig. 3b, our model surpasses UNICOM on 23 datasets and achieves an average improvement of 1.3%, confirming the effectiveness of the proposed multi-label loss.

### 4.3 ZERO-SHOT CLASSIFICATION

In Tab. 2, we present a comparison of our method with state-of-the-art approaches in zero-shot classification on 25 datasets. The prompt templates and class names are consistent with previous works (Li et al., 2023b). As depicted in Fig. 3c, our method surpasses OpenCLIP on 23 datasets with 3.9% average performance improvement. Although FLIP uses masking to save memory footprint to learn more samples per iteration, our method demonstrates better results on 15 out of 25 datasets in Fig. 3d, and achieves a significant performance boost of 1.5% on average.

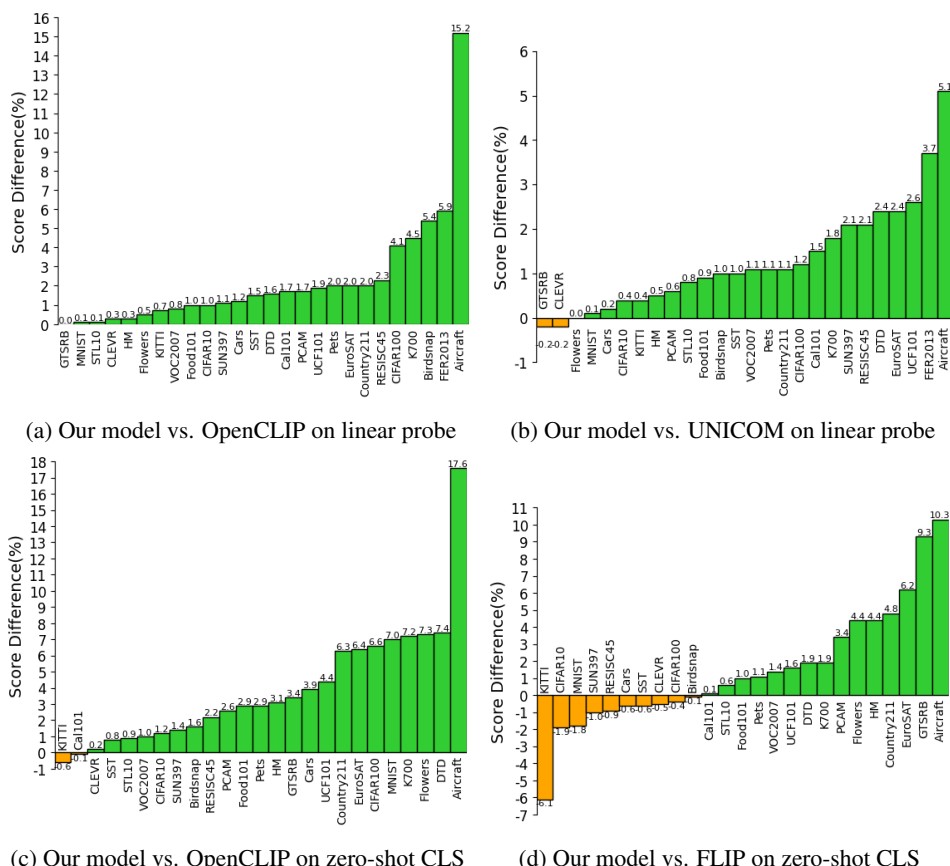

(a) Our model vs. OpenCLIP on linear probe    (b) Our model vs. UNICOM on linear probe

(c) Our model vs. OpenCLIP on zero-shot CLS    (d) Our model vs. FLIP on zero-shot CLS

Figure 3: Linear probe and zero-shot comparisons on different downstream datasets. Y-axis shows the performance difference. Green bars indicate our model outperforms the baselines, while the orange bars depict our model is surpassed by the baselines.

## 4.4 ZERO-SHOT RETRIEVAL

Tab. 3 reports zero-shot image-text retrieval results on Flickr30k and MSCOCO. In comparison to OpenCLIP, our model achieves 60.8%/44.5% I2T/T2I retrieval Recall@1 on the MSCOCO dataset, which is 2.8%/3.2% higher than OpenCLIP. Similarly, our model demonstrates significant improvements of 1.8%/3.9% on the Flickr30k dataset. Furthermore, compared to FLIP, our model exhibits either competitive or superior retrieval performance.

| | | Text retrieval | | | | | | Image retrieval | | | | | |
| | | Flickr30k | | | MSCOCO | | | Flickr30k | | | MSCOCO | | |
| CASE | DATA | R@1 | R@5 | R@10 | R@1 | R@5 | R@10 | R@1 | R@5 | R@10 | R@1 | R@5 | R@10 |
| --- | --- | --- | --- | --- | --- | --- | --- | --- | --- | --- | --- | --- | --- |
| CLIP[‡] | WIT-400M | 87.8 | 99.1 | 99.8 | 56.2 | 79.8 | 86.4 | 69.3 | 90.2 | 94.0 | 35.8 | 60.7 | 70.7 |
| OpenCLIP[‡] | LAION-400M | 87.3 | 97.9 | 99.1 | 58.0 | 80.6 | 88.1 | 72.0 | 90.8 | 95.0 | 41.3 | 66.6 | 76.1 |
| FLIP[‡] | LAION-400M | 89.1 | 98.5 | 99.6 | 60.2 | 82.6 | 89.9 | 75.4 | 92.5 | 95.9 | 44.2 | 69.2 | 78.4 |
| Ours | LAION-400M | 89.1 | 98.4 | 99.5 | 60.8 | 83.2 | 91.3 | 75.9 | 93.1 | 96.8 | 44.5 | 69.6 | 79.9 |

Table 3: Zero-shot image-text retrieval on the test splits of Flickr30k and MSCOCO. ‡: Results reported in FLIP paper. Entries in green are the best results using LAION-400M.

## 4.5 ZERO-SHOT ROBUSTNESS EVALUATION

Following FLIP (Li et al., 2023b), we conduct a robustness evaluation as shown in Tab. 4. In comparison to the models pre-trained on LAION, our method demonstrates superior robustness compared to both OpenCLIP and FLIP. It is worth noting that the performance gap between our model pre-trained on LAION and CLIP pre-trained on WIT arises from the statistical differences in pre-training data.

| CASE | DATA | IN-V2 | IN-A | IN-R | ObjectNet | IN-Sketch |
|------|------|-------|------|------|-----------|-----------|
| CLIP[‡] | WIT-400M | 69.5 | 71.9 | 86.8 | 68.6 | 58.5 |
| OpenCLIP[‡] | LAION-400M | 64.0 | 48.3 | 84.3 | 58.8 | 56.9 |
| FLIP[‡] | LAION-400M | 66.8 | 51.2 | 86.5 | 59.1 | 59.9 |
| Ours | LAION-400M | 68.9 | 56.4 | 85.1 | 62.7 | 60.4 |

Table 4: Zero-shot robustness evaluation. ‡: Results reported in FLIP paper. Entries in green are the best results using LAION-400M. Here, all methods employ the model backbone of ViT-L/14.

## 4.6 IMAGENET CLASSIFICATION

We evaluate performance on ImageNet (Deng et al., 2009) under three distinct settings: finetuning, linear classification, and zero-shot. As shown in Tab. 5, our ViT-L/14 model achieves better performance on all settings, indicating that multi-label cluster discrimination can better encode the semantics of data than instance discrimination and cluster discrimination.

| CASE | DATA | Finetune | Linear Probe | Zero Shot |
|------|------|----------|--------------|-----------|
| CLIP | WIT-400M | - | 83.9 | 75.3 |
| OpenCLIP[‡] | LAION-400M | 86.2 | 82.1 | 72.8 |
| FLIP[‡] | LAION-400M | - | - | 74.6 |
| UNICOM | LAION-400M | - | 81.8 | - |
| Ours | LAION-400M | 87.1 | 84.6 | 75.6 |

Table 5: ImageNet results under finetuning, linear probe, and zero-shot settings. ‡: Results reported in FLIP paper. Here, all methods employ the model backbone of ViT-L/14.

## 4.7 ABLATION STUDY

In Tab. 6, we first conduct ablation experiments to investigate the impact of hyper-parameters. In Tab. 7, we verify the effectiveness of the proposed contrastive loss decomposition and the efficacy of our multi-label learning on ImageNet.

**Number of Classes.** The number of classes ($k$) plays a crucial role in balancing inter-class conflict and intra-class purity. In Tab. 6a, we observe that as the number of classes increases from 10K to 1M, there is a gradual increase in intra-class purity, leading to an improved performance on ImageNet. However, as the number of classes continues to increase from 1M to 5M, inter-class conflicts gradually escalate, resulting in a deteriorated performance.

**Sample Ratio.** The sample ratio ($r$) influences the number of negative samples and directly affects the likelihood of encountering inter-class conflicts. A sample ratio of 0.01 yields a linear probe performance of only 73.4% due to the limited number of negative samples, which adversely affects the representation learning. Conversely, a sample ratio of 1.0 substantially increases the probability of encountering inter-class conflicts. Tab. 6b presents that the superior linear probe performance of 75.2% is achieved when employing a sample ratio of 0.1.

| # classes | 10K | 20K | 50K | 1M | 2M | 5M |
|-----------|-----|-----|-----|-----|-----|-----|
| IN1K | 66.9 | 71.1 | 74.4 | 75.2 | 74.9 | 74.7 |

(a) The number of **classes** in training set.

| sample ratio | 0.01 | 0.05 | 0.1 | 0.2 | 0.5 | 1.0 |
|--------------|------|------|-----|-----|-----|-----|
| IN1K | 73.4 | 75.1 | 75.2 | 74.9 | 68.3 | 63.2 |

(b) The **ratio** of sampled negative class centers.

| positive centers | 1 | 2 | 4 | 8 | 16 | 32 |
|------------------|---|---|---|---|----|----|
| IN1K | 71.4 | 72.9 | 73.2 | 75.2 | 72.1 | 68.7 |

(c) The effect of **multi labels** per sample.

| positive threshold | 0.95 | 0.93 | 0.91 | 0.89 | 0.87 | 0.85 |
|--------------------|------|------|------|------|------|------|
| IN1K | 72.2 | 72.7 | 73.3 | 72.4 | 68.7 | 63.2 |

(d) The effect of varying **positive thresholds**.

Table 6: **Ablation experiments**. The model backbone used here is ViT-B/32. Pre-training is executed on the LAION-400M dataset for a duration of 5 epochs. Performance assessment is undertaken using a linear probe on the ImageNet validation set.

| CASE | DATA | Finetune | Linear Probe | Zero Shot |
|------|------|----------|--------------|-----------|
| MLC | LAION-400M | 80.9 | 76.9 | 63.9 |
| MLCD | LAION-400M | 81.2 | 78.1 | 64.5 |

(a) Efficacy of contrastive loss decomposition.

| CASE | DATA | 0.5K | 1K | 2K | 4K | 8K | 20K |
|------|------|------|-----|-----|-----|-----|-----|
| UNICOM | IN1K | 42.1 | 58.4 | 61.5 | 62.8 | 62.4 | 61.5 |
| MLCD | IN1K | 63.2 | 67.2 | 68.2 | 69.9 | 69.7 | 69.0 |

(b) Efficacy of multi-label learning on ImageNet.

Table 7: **Ablation experiments of (a) the proposed contrastive loss decomposition and (b) the multi-label learning on ImageNet**. For (a), pre-training is executed on the LAION-400M dataset by 32 epochs. The model backbone used here is ViT-B/32. Results are reported on the ImageNet validation dataset. For (b), pre-training is executed on the ImageNet training dataset by 100 epochs. The model backbone used here is ResNet-50. The evaluation is undertaken using a linear probe on the ImageNet validation set.

**Multi-label Assignment.** We explore two different approaches to obtain multi-labels. Firstly, we artificially assign a predetermined number of labels to each sample. Tab. 6c presents linear probe results on ImageNet with different numbers of positive centers. Consequently, we observe a gradual improvement in performance as the number of positive centers increases from 1 to 8. However, as the number of positive centers continues to increase, the inclusion of excessive positive centers introduces noise labels, leading to a degradation in performance. Additionally, we have also investigated the use of sample-cluster similarity thresholds to obtain multiple labels. This approach results in varying numbers of positive centers associated with each sample. However, as shown in Tab. 6d, the performance of applying adaptive positive centers is generally lower compared to that of using fixed assignment of positive centers (Tab. 6c). This indicates that the global similarity threshold is hard to search while the fixed assignment strategy benefits from the prior that the daily image statistically contains several visual concepts.

**Effectiveness of Contrastive Loss Decomposition.** In Tab. 7a, we compare the performance of the vanilla MLC (Eq. 4) and the proposed MLCD (Eq. 5) on the ImageNet. Both MLC and MLCD employ the negative class center sampling with a ratio of $0.1$. MLCD outperforms MLC in all three settings: fine-tuning, linear classification, and zero-shot, confirming the effectiveness of the two additional optimization targets. In Appendix. A.1, we compare their gradient calculation and time cost on the classification layer. The proposed contrastive loss decomposition can significantly decrease the communication cost, facilitating distributed training on large-scale training data.

**Effectiveness of Multi-label Learning on ImageNet.** In Tab. 7b, we compare the proposed multi-label cluster discrimination and the single-label cluster discrimination on ImageNet with the clustered class number ranging from 0.5K to 20K. The clustering step is conducted by using the features predicted by the CLIP model. In the discrimination step, both MLCD and UNICOM employ the negative class center sampling with a ratio of $0.1$, and the positive number for MLCD is set as $8$. As we can see, the proposed multi-label learning significantly surpasses UNICOM and achieves the best performance of $69.9\%$ when the class number is 4K, which is four times of the true class number of ImageNet. In Fig. 5 of the Appendix, we visualize the top three labels for our training samples. When training with multiple labels, our method can learn complementary visual signals (e.g., different breeds of dogs, different locations of figs) to improve visual representation learning.

## 5 DISCUSSION AND CONCLUSION

In this paper, we propose a novel multi-label cluster discrimination method to cope with multiple visual signals existing in one image. Compared to the vanilla version of the multi-label loss (i.e., seeking to narrow the gap between the inter-class similarities and intra-class similarities), our methods enable elegant separation of losses from positive and negative classes. Extensive experimental results show that the proposed multi-label loss is effective for providing better transferrable features on multiple downstream tasks than both instance and cluster discrimination methods.

**Limitations.** We use the publicly available large-scale training data (i.e., the LAION-400M dataset) for training, and the resulting model weights reflect the intrinsic data biases, including potentially negative implications. For example, our model is less competitive in activity classification compared to the CLIP model trained on the private WIT-400M dataset.

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

# A APPENDIX

## A.1 GRADIENT CALCULATION OF MLC AND MLCD

For the baseline multi-label classification method (Eq. 4), the gradients with respect to $s_i$ and $s_j$ are derived as follows:

$$
\begin{aligned}
\frac{d\mathcal{L}_{\mathrm{MLC}}}{ds_i} &= \frac{-\exp(-s_i) \cdot \sum_{j \in \Omega_n} \exp(s_j)}{1 + \sum_{j \in \Omega_n} \exp(s_j) \cdot \sum_{i \in \Omega_p} \exp(-s_i)} \\
&= \frac{-\exp(-s_i) \cdot \mathrm{allreduce}(\mathbf{1}^\top \exp(s_j))}{1 + \mathrm{allreduce}(\mathbf{1}^\top \exp(s_j) \cdot \mathrm{allreduce}(\mathbf{1}^\top \exp(-s_i)))},
\end{aligned}
\tag{7}
$$

and

$$
\begin{aligned}
\frac{d\mathcal{L}_{\mathrm{MLC}}}{ds_j} &= \frac{\exp(s_j) \cdot \sum_{j \in \Omega_n} \exp(-s_i)}{1 + \sum_{j \in \Omega_n} \exp(s_j) \cdot \sum_{i \in \Omega_p} \exp(-s_i)} \\
&= \frac{\exp(s_j) \cdot \mathrm{allreduce}(\mathbf{1}^\top \exp(-s_i))}{1 + \mathrm{allreduce}(\mathbf{1}^\top \exp(s_j) \cdot \mathrm{allreduce}(\mathbf{1}^\top \exp(-s_i)))},
\end{aligned}
\tag{8}
$$

where $\mathrm{allreduce}(\cdot)$ denotes a parallel reduction operation that collects the sum of the exponentials from all processes and then redistributes the calculated result back to each process. For the proposed multi-label cluster discrimination method (Eq. 5), the gradients of the positive similarity score $s_i$ and the negative similarity score $s_j$ are derived as follows:

$$
\frac{d\mathcal{L}_{\mathrm{MLCD}}}{ds_i} = \frac{-\exp(-s_i)}{1 + \sum_{i \in \Omega_p} \exp(-s_i)} = \frac{-\exp(-s_i)}{1 + \mathrm{allreduce}\left(\mathbf{1}^\top \exp(-s_i)\right)},
\tag{9}
$$

$$
\frac{d\mathcal{L}_{\mathrm{MLCD}}}{ds_j} = \frac{\exp(s_j)}{1 + \sum_{j \in \Omega_n} \exp(s_j)} = \frac{\exp(s_j)}{1 + \mathrm{allreduce}\left(\mathbf{1}^\top \exp(s_j)\right)}.
\tag{10}
$$

The above formulas illustrate the derivatives of the loss function $\mathcal{L}_{\mathrm{MLC}}$ and $\mathcal{L}_{\mathrm{MLCD}}$ with respect to the similarity scores. As we can see, the proposed method enables an elegant separation of positive and negative gradient calculation, which can decrease the communication frequency of calling the $\mathrm{allreduce}(\cdot)$ operation. To compare the time cost on the classification layer, we train UNICOM (An et al., 2023), $\mathcal{L}_{\mathrm{MLC}}$, and $\mathcal{L}_{\mathrm{MLCD}}$ on the LAION-400M dataset with one million classes. We use ViT-B/32 as the backbone, and the final embedding dimension is 512. The batch size is set as $32,800$ and the A100 GPU number is 80 in total distributed across 10 computation nodes. The time cost of the forward and backward steps regarding the backbone is 409ms. For $\mathcal{L}_{\mathrm{MLC}}$ and $\mathcal{L}_{\mathrm{MLCD}}$, the positive class number is set as 8. For the single-label classification method, UNICOM, the time cost on the classification layer is 75ms. For $\mathcal{L}_{\mathrm{MLC}}$ and $\mathcal{L}_{\mathrm{MLCD}}$, the time cost on the classification layer is 138ms and 82ms. The proposed multi-label cluster discrimination method slightly increases the time cost by 9.3% on the classification layer compared to the single-label cluster discrimination method (An et al., 2023). Compared to the widely used multi-label classification method, the proposed method obviously decreases the time cost by 45.6% on the classification layer. Therefore, the proposed decomposition of contrastive loss ensures efficient parallel computation across multiple GPUs from different computation nodes with minimal communication overhead.

## A.2 PRE-TRAINING DETAILS

**Encoders.** Tab. 8 shows the architectures we use. The designs follow CLIP Radford et al. (2021). Our image encoders involve ViT-B and ViT-L, using the same patch size as in CLIP.

| Model | Learning rate | Embedding dimension | Input resolution | Vision Transformer | | | Text Transformer | | |
|---|---|---|---|---|---|---|---|---|---|
| | | | | layers | width | heads | layers | width | heads |
| ViT-B/32 | $5 \times 10^{-4}$ | 512 | 224 | 12 | 768 | 12 | 12 | 512 | 8 |
| ViT-B/16 | $5 \times 10^{-4}$ | 512 | 224 | 12 | 768 | 12 | 12 | 512 | 8 |
| ViT-L/14 | $4 \times 10^{-4}$ | 768 | 224 | 24 | 1024 | 16 | 12 | 768 | 12 |

Table 8: ViT hyper-parameters.

**Hyper-parameters.** Our default pre-training configuration is shown in Tab. 9. During the training process of the text encoder, the hyperparameters are the same as those of the pre-training for the image encoder. The vision model is frozen, preventing any backpropagation of gradients. When calculating the multi-label contrastive loss, we follow the approaches of ArcFace (Deng et al., 2019) and Unicom (An et al., 2023), we apply L2 normalization to both the features and the class centers, and introduce a margin ($m = 0.3$) for the positive classes.

| Hyperparameter | Value |
|---|---|
| Batch size | 32800 |
| Vocabulary size | 49408 |
| Training epochs | 32 |
| Maximum temperature | 100.0 |
| Weight decay | 0.2 |
| Warm-up iterations | 2000 |

Table 9: Training hyperparameters.

**Scalability.** In Fig. 4a and Fig. 4b, we validate the scalability of our method. Scaling up the ViT model and incorporating more data both significantly enhance our model's performance.

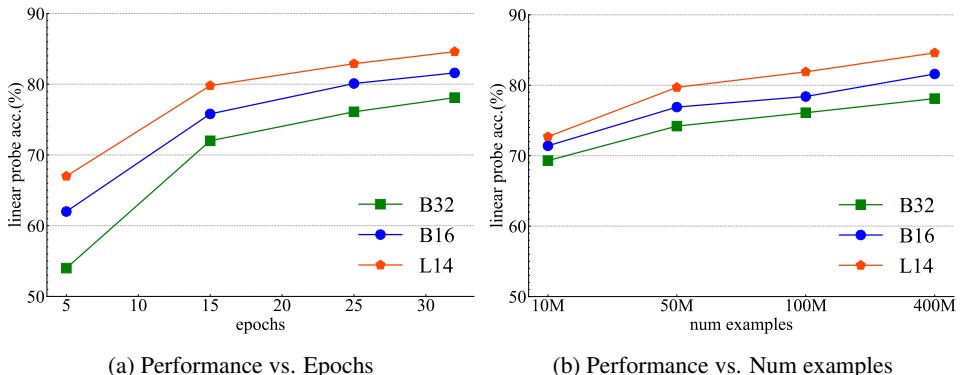

(a) Performance vs. Epochs      (b) Performance vs. Num examples

Figure 4: (a) the convergence curves of different ViTs. (b) the scalability curves of different ViTs under varying dataset scales. Larger ViTs and datasets lead to better model performance.

### A.3 LINEAR PROBE EVALUATION

In our linear probing analysis, we adhered to the same configuration as employed by CLIP. We utilized the L-BFGS optimization algorithm as implemented in PyTorch, executing it on a GPU with an upper limit of 1000 iterations. We adopted CLIP's parametric binary search protocol to optimize the hyperparameter $\lambda$, with the optimization process conducted on the validation set. In cases where a dataset lacks a predefined validation set, we manually partition the dataset. This streamlined methodology allowed us to efficiently run tests across all 26 datasets within a few hours. For the final results, the validation set is merged back into the training set for an additional round of training.

### A.4 ZERO-SHOT EVALUATION

For the experiments in Tab. 2, we use the prompts same as FLIP. Following CLIP (Radford et al., 2021), we report the mean accuracy per class for FGVC Aircraft, Oxford-IIIT Pets, Caltech-101, and Oxford Flowers 102 datasets. We report the mean of top-1 and top-5 accuracy for Kinetics-700, ROC AUC for Hateful Memes, and 11-point mAP for Pascal VOC 2007 Classification. We report top-1 accuracy for the rest of the datasets.

| Dataset | Classes | Train size | Test size | Evaluation metric |
|---|---|---|---|---|
| Food101 | 102 | 75,750 | 25,250 | accuracy |
| CIFAR10 | 10 | 50,000 | 10,000 | accuracy |
| CIFAR100 | 100 | 50,000 | 10,000 | accuracy |
| Birdsnap | 500 | 42,138 | 2,149 | accuracy |
| SUN397 | 397 | 19,850 | 19,850 | accuracy |
| Cars | 196 | 8,144 | 8,041 | accuracy |
| Aircraft | 100 | 6,667 | 3,333 | mean per class |
| VOC2007 | 20 | 5011 | 4952 | 11-point mAP |
| DTD | 47 | 3,760 | 1,880 | accuracy |
| Pets | 37 | 3,680 | 3,669 | mean per class |
| Caltech101 | 101 | 3,000 | 5,677 | mean-per-class |
| Flowers | 102 | 2,040 | 6,149 | mean per class |
| MNIST | 10 | 60,000 | 10,000 | accuracy |
| STL10 | 10 | 5,000 | 8,000 | accuracy |
| EuroSAT | 10 | 10,000 | 5,000 | accuracy |
| RESISC45 | 45 | 3,150 | 25,200 | accuracy |
| GTSRB | 43 | 26,640 | 12,630 | accuracy |
| KITTI | 4 | 6770 | 711 | accuracy |
| Country211 | 211 | 42,200 | 21,100 | accuracy |
| PCAM | 2 | 294,912 | 32,768 | accuracy |
| UCF101 | 101 | 9,537 | 1,794 | accuracy |
| Kinetics700 | 700 | 530,779 | 33,944 | mean(top1,top5) |
| CLEVR | 8 | 2,000 | 500 | accuracy |
| Memes | 2 | 8,500 | 500 | ROC AUC |
| SST2 | 2 | 7,792 | 1,821 | accuracy |
| ImageNet | 1000 | 1,281,167 | 50,000 | accuracy |

Table 10: List of linear probe datasets with the data distribution and evaluation metrics.

## A.5 ZERO-SHOT RETRIEVAL

We assess the effectiveness of zero-shot retrieval using two established benchmarks: Flickr30K (Young et al., 2014) and COCO (Lin et al., 2014), each containing 1K and 5K image-text pairs in their test sets, respectively. In adhering to the procedures outlined in CLIP and FLIP, we derive the image and text embeddings from the relevant encoders, and then execute retrieval by calculating cosine similarities across potential image-text pairs, without prompt being utilized.

## A.6 ZERO-SHOT ROBUSTNESS EVALUATION

In our zero-shot robustness assessment on ImageNet related sets, we employ the 7 prompts provided by CLIP, with dataset preparation and division adhering to the methods used in OPENCLIP. For ObjectNet, we emulate the approach of CLIP by utilizing class names without any prompt.

## A.7 DOWNSTREAM DATASETS

We use 26 image classification datasets to prove the effectiveness of our method. These datasets include Food101 Bossard et al. (2014), CIFAR10 Krizhevsky et al. (2009), CIFAR100 Krizhevsky et al. (2009), Birdsnap Berg et al. (2014), SUN397 Xiao et al. (2010), Stanford Cars Krause et al. (2013), FGVC Aircraft Maji et al. (2013), VOC2007 Everingham (2007), DTD Cimpoi et al. (2014), Pets Parkhi et al. (2012), Caltech101 Fei-Fei et al. (2004), Flowers102 Nilsback & Zisserman (2008), MNIST LeCun et al. (1998), SLT10 Coates et al. (2011), EuroSAT Helber et al. (2019), RESISC45 Cheng et al. (2017), GTSRB Stallkamp et al. (2012), KITTI Geiger et al. (2012), Country211 Radford et al. (2021), PCAM Veeling et al. (2018), UCF101 Soomro et al. (2012), Kinetics700 Carreira et al. (2019), CLEVR Johnson et al. (2017), Hateful Memes Kiela et al. (2020), SST2 Radford et al. (2021), ImageNet Deng et al. (2009).Details on each dataset and the corresponding evaluation metrics are provided in Tab. 10.

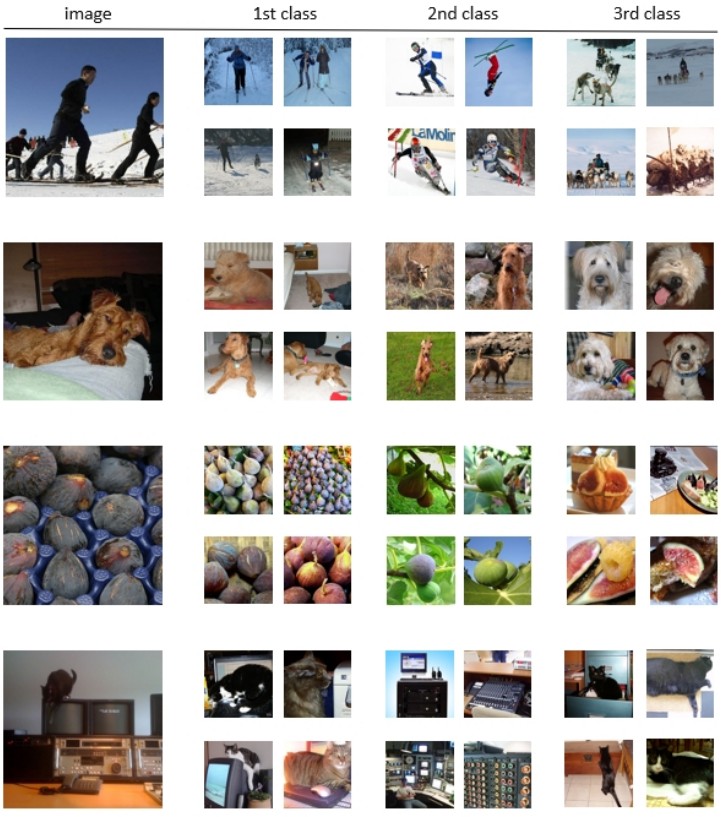

Figure 5: Visualization of top 3 labels given to the training samples from the automatically clustered ImageNet dataset. Multiple positive labels show complementary visual signals.

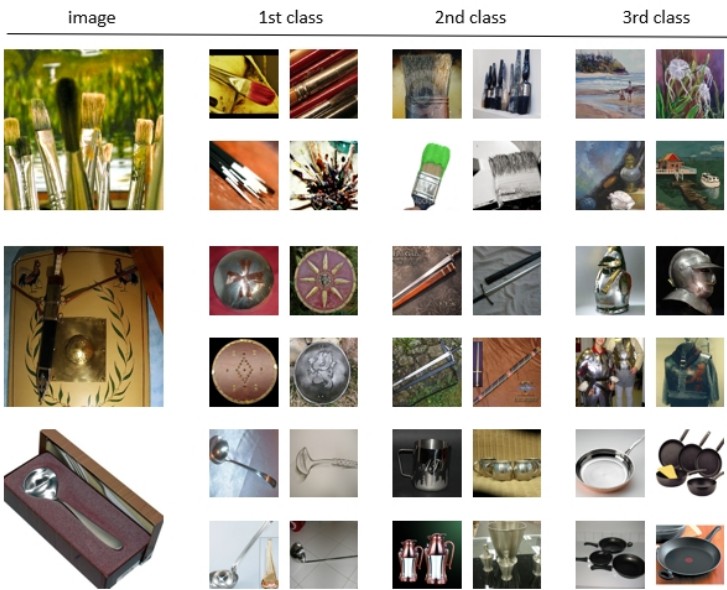

Figure 6: Some interesting labels given to the training samples from the automatically clustered ImageNet dataset. Multiple positive labels sometimes show abstract correlations, e.g., the concept of "colorful" between "paintbrush" and "oil painting" in the 1st row, the concept of "shining" between "shield" and "armour" in the 2nd row, the concept of "with a handle" between "ladle" and "frying pan" in the 3rd row.

### A.8 MULTI-LABEL LEARNING ON IMAGENET

In Fig. 5, we visualize the top 3 labels given to some selected training samples. We cluster the original ImageNet training dataset into 4K classes by using the pre-trained CLIP model. As depicted in Fig. 5, these multiple positive labels exhibit complementary visual signals (e.g., different activities on the snow, different breeds of dogs, different locations of figs, and different objects in the room). For each of the automatically clustered 4K classes, we match the dominant class to the original 1K classes. In the single-label cluster discrimination method, only one label is used for training, and the recall of the correct label is $76.2\%$. By contrast, in the proposed multi-label cluster discrimination, the top 8 labels are used for training, and the recall of the correct label is $93.5\%$. Even though using multiple labels introduces some incorrect labels, the proposed multi-label cluster discrimination method can learn complementary visual signals to improve visual representation learning.

In Fig. 6, we also visualize some interesting labels given to the automatically clustered samples. As we can see, multiple positive labels sometimes show abstract correlations. For example, there is the concept of "colorful" between "paintbrush" and "oil painting" in the 1st row, the concept of "shining" between "shield" and "armour" in the 2nd row, and the concept of "with a handle" between "ladle" and "frying pan" in the 3rd row.

### A.9 BROADER IMPACTS

Training the large-scale vision model requires high energy consumption thus resulting in large amounts of carbon emissions. Even though our multi-label cluster discrimination method only employs one-step offline clustering, the remaining cost in the multi-label classification step is still sizable.

