# OpenReview forum: "Multi-label Cluster Discrimination for Visual Representation Learning"
_ICLR.cc/2024/Conference — Submitted to ICLR 2024_

### Official Review · Reviewer_KbPe · 2023-10-24

**Soundness:** 3 good
**Presentation:** 3 good
**Contribution:** 3 good
**Rating:** 5
**Confidence:** 3

**Summary:**

This work focuses on addressing the limitations of instance discrimination commonly used in image-text contrastive learning, such as CLIP. The authors propose a multi-label cluster discrimination method aimed at improving the encoding ability. They employ offline clustering to assign multiple labels to each image and subsequently conduct multi-label classification to learn the semantic structure within a single image. The authors support their methods with extensive experiments and perform ablation studies to analyze the function of each component.

**Strengths:**

1. This work considers the multi-label properties of a single image and emphasizes the learning of better semantic structure in data.
2. The designed loss function elegantly separates the loss from positive and negative classes, which enhances the parallelism and scalability during training.
3. The experiments in this work are extensive and convincing, with thorough ablation studies.

**Weaknesses:**

1. Clarity:
   - This manuscript requires further refinement in terms of writing to facilitate reader comprehension, particularly by providing detailed explanations for the mathematical symbols used in the text, thus reducing reading barriers.
2. Experiments:
   - In section $3.2, the authors claim efficient parallel computation and scalability of the model training process. However, is there quantitative data to support this point?
   - Does the incorporation of clustering significantly improve the training time?
3. Reproducibility:
   - In section $3.2, you employ some distribution training techniques but details are not provided, which hinders the reproducibility of the work.

**Questions:**

1. The performance reported in the CLIP paper differs from your reproduced version. In Tab 1 and Tab 2, which may influence the validation of improvement of your model. Have you checked the implementation and settings?

---

> ### Author Response · Authors · 2023-11-23
> **Response to Reviewer KbPe**
>
> Thanks for your valuable suggestions to improve our paper. We were glad to see your positive ranking at the beginning. Sorry for our late reply due to the time cost of additional experiments. We hope the revised version has solved all of your concerns on this paper.
>
> **Weakness1: detailed explanations for the mathematical symbols used in the text .**
>
> **A1:**  In the revised version, we add detailed explanations for all mathematical symbols used in the equations and text.
>
> **Weakness2: In section 3.2, the authors claim efficient parallel computation and scalability of the model training process. However, is there quantitative data to support this point? Does the incorporation of clustering significantly improve the training time?**
>
> **A2:**  In the revised Appendix A.1, we add a detailed explanation regarding the gradient calculation and we also include time cost comparison on the classification layer.
>
> As we can see, the proposed method enables an elegant separation of positive and negative gradient calculation, which can decrease the communication frequency of calling the allreduce operation.
> To compare the time cost on the classification layer, we train UNICOM, MLC, and MLCD on the LAION-400M dataset with one million classes.
> We use ViT-B/32 as the backbone, and the final embedding dimension is 512.
> The batch size is set as $32,800$ and the A100 GPU number is 80 in total distributed across 10 computation nodes.
> The time cost of the forward and backward steps regarding the backbone is $409$ms.
> For MLC and MLCD, the positive class number is set as 8.
> For the single-label classification method, UNICOM, the time cost on the classification layer is $75$ms.
> For MLC and MLCD, the time cost on the classification layer is $138$ms and $82$ms. The proposed multi-label cluster discrimination method slightly increases the time cost by 9.3% on the classification layer compared to the single-label cluster discrimination method, UNICOM. Compared to the widely used multi-label classification method, the proposed method obviously decreases the time cost by 45.6% on the classification layer. Therefore, the proposed decomposition of contrastive loss ensures efficient parallel computation across multiple GPUs from different computation nodes with minimal communication overhead.
>
> With the aid of efficient feature quantization (Johnson et al., 2019), it only takes around 10 minutes to complete the offline clustering step on the large-scale LAION-400M dataset. For multi-label training, there is a slight increase of 9.3% in the time cost of the classification layer compared to the single-label cluster discrimination method.
>
> **Weakness3:  In section 3.2, you employ some distribution training techniques but details are not provided, which hinders the reproducibility of the work.**
>
> **A3:**  In the revised Appendix A.1, we add a detailed explanation regarding the gradient calculation and the distribution training. In the supplementary material, we put our training code, which is visible to the public now and can be used for reproducing our method.
>
> **Q1: The performance reported in the CLIP paper differs from your reproduced version. In Tab 1 and Tab 2, which may influence the validation of improvement of your model. Have you checked the implementation and settings?**
>
> **A1:** Even though CLIP has generously open-sourced the models, they have not provided the evaluation tools.
> To this end, we contacted the UNICOM [1] authors to get their evaluation toolkit and all test datasets for the task of linear probe. In UNICOM, the authors have developed a GPU-accelerated logistic regression algorithm, facilitating batch-wise assessment of linear probe performance. In this paper, the performance of CLIP is also reported by using this evaluation toolkit.
>
> For the task of zero-shot classification, we adopted the same experimental settings as FLIP [2] and referred to the FLIP paper to report the zero-shot results.
>
> [1] An, Xiang, et al. Unicom: Universal and Compact Representation Learning for Image Retrieval. ICLR 2023.
>
> [2] Li, Yanghao, et al. Scaling language-image pre-training via masking. CVPR 2023.

---

### Official Review · Reviewer_g3TE · 2023-10-30

**Soundness:** 3 good
**Presentation:** 2 fair
**Contribution:** 3 good
**Rating:** 5
**Confidence:** 3

**Summary:**

This paper proposes a new clustering-based unsupervised algorithm for vision foundation models pre-training. The key idea is to assign images into multiple clusters as pseudo labels for unsupervised representation learning. The motivation is that existing clustering-based pre-training methods assign each image into a single cluster, which enforces the models to focus on the most salient part of images and overlook the other regions that may also be meaningful. Besides, the authors also optimise the conventional margin loss formulation by decoupling the optimisations of positive and negative pairwise similarity. The proposed algorithm has been shown effective in severe classification-oriented downstream tasks including linear probe, zero-shot classification and retrieval.

**Strengths:**

The algorithm proposed in this paper is intuitive and effective in learning discriminative imagery feature representations. The analysis and decomposition of triplet loss make sense to me and are potentially beneficial to a wide range of applications as a generic improvement to a widely adopted metric learning design.

**Weaknesses:**

+ my key concern on the high-level idea is whether the top-k closest clusters to an image can really reveal what objects/attributes (will use “concepts” for clarity hereafter) are involved in it. At the cluster level, samples of the same clusters are likely to share more nearest clusters (in a global picture) but the concepts involved in each independent image are almost random. Is it possible that the multiple labels assigned to the same images provide models with additional knowledge about the co-occurrence/relevance of different concepts (cluster-to-cluster relationships) rather than actually telling models what is involved in images (sample-to-cluster relationships)? It will be interesting to see more exploration and analysis of why the multi-label clustering idea is beneficial. One simple verification can be pre-training on a dataset with known non-overlapping class structure, eg ImageNet, and see if the multi-label clustering still benefits.

+ The modifications made to triplet loss make sense to me but their effects are unclear. How will the proposed model perform if all its designs are kept unchanged except for replacing L’_MLCD (Eq.6) with L_MLC


+ What are the blue and green cells standing for in the grids pointing to the text “contrastive loss”?

+ Whilst Fig.2 is the first figure being referred to, Fig.1 is simply mentioned as the illustration of visual representation learning but it lacks further explanation/discussion.

+ In Eq.1, I assume the pairwise similarity is cosine similarity if following CLIP, but without normalisation of features, it is just an inner product. So I’m wondering if it is a mistake or my misunderstanding.

+ In Eq.1, the index in the cumulative sum starts from 0 to k while that in Eq.2 is from 1 to k, is this deliberate and why?

+ The exponential function is denoted as exp and e at the same time in Eq.3, which makes the equation really confusing when the feature representations are also denoted as e_i.

+ The ablation studies are a bit unclear to me. For example, when investigating the effects of sample ratio, the best linear probe performance is obtained when the sample ratio is set to 0.1, and the best result is 75.2. However, the linear probe performance of the proposed model shown in Table 5 is 84.6.

**Questions:**

Although the proposed method yielded impressive performance, it is also crucial for me to figure out the underlying reasons for the effectiveness. So further evidence and discussions about this will be helpful.

---

> ### Author Response · Authors · 2023-11-23
> **Response to Reviewer g3TE**
>
> Thanks for your insightful suggestions to improve our paper.
>
> **Q1: Is it possible that the multiple labels assigned to the same images provide models with additional knowledge about the co-occurrence/relevance of different concepts (cluster-to-cluster relationships) rather than actually telling models what is involved in images (sample-to-cluster relationships)? pre-training on a dataset with known non-overlapping class structure, eg ImageNet, and see if the multi-label clustering still benefits.**
>
> **A1:** We added corresponding experiments and visualization in the revised version.
>
>  |   CASE   | DATA | 0.5K | 1K   | 2K   | 4K   | 8K   | 20K |
>  | ------   | ------ | ------ | ------ | ------ | ------ | ------ | ------ |
>  |   UNICOM | IN1K | 42.1 | 58.4 | 61.5 | 62.8 | 62.4 | 61.5|
>  |   MLCD   | IN1K | 63.2 | 67.2 | 68.2 | **69.9** | 69.7 | 69.0|
>
> In Tab. 7 (a), we compare the proposed multi-label cluster discrimination and the single-label cluster discrimination on ImageNet with the clustered class number ranging from 0.5K to 20K. The clustering step is conducted by using the features predicted by the CLIP model. In the discrimination step, both MLCD and UNICOM employ the negative class center sampling with a ratio of $0.1$, and the positive number for MLCD is set as $8$.
> Pre-training is executed on the ImageNet training dataset by 100 epochs. The model backbone is ResNet-50. The evaluation is undertaken using a linear probe on the ImageNet validation set.
> As we can see, the proposed multi-label learning significantly surpasses UNICOM and achieves the best performance of 69.9% when the class number is 4K, which is four times of the true class number of ImageNet. In Fig. 5 of the Appendix, we visualize the top three labels for our training samples. When training with multiple labels, our method can not only capture the label correlations (e.g., paintbrush and oil painting, shield and armour, ladle and frying pan) but also learn complementary visual signals (e.g., different breeds of dogs, different locations of figs) to improve visual representation learning.
>
> **Q2: The modifications made to triplet loss make sense to me but their effects are unclear. How will the proposed model perform if all its designs are kept unchanged except for replacing L’_MLCD (Eq.6) with L_MLC.**
>
> **A2:** In the revised version, we add the experiments to verify the effectiveness of the proposed MLCD.
>
> |CASE | DATA | Finetune | Linear Probe | Zero Shot|
> | ------   | ------ | ------ | ------ | ------ |
> |MLC  | LAION-400M| 80.9 | 76.9 | 63.9 |
> |MLCD | LAION-400M| 81.2 | 78.1 | 64.5 |
>
> Tab. 7 (a), we compare the performance of the vanilla MLC and the proposed MLCD on the ImageNet validation dataset. Pre-training is executed on the LAION-400M dataset by 32 epochs. The model backbone is ViT-B/32.
> Both MLC and MLCD employ the negative class center sampling with a ratio of $0.1$. MLCD outperforms MLC in all three settings: fine-tuning, linear classification, and zero-shot, confirming the effectiveness of the two additional optimization targets. In Appendix. A.1, we compare their gradient calculation and time cost on the classification layer. The proposed contrastive loss decomposition can significantly decrease the communication cost, facilitating distributed training on large-scale training data.
>
> **Q3: What are the blue and green cells standing for?**
>
> **A3:**  For Fig. 1(a), blue cells stand for the similarity scores of positive image-text pairs. The similarities in the blue cells need to be increased during training.
> For Fig. 1(b), there are two training samples in the minibatch.
> The blue and green cells stand for the positive scores in their corresponding one-hot target logits.
> For Fig. 1(c), there are also two training samples in the minibatch.
> However, each sample has three positive labels. Thus, the target logits contain three positive similarity scores that need to be increased during training.
>
> **Q4: Whilst Fig.2 is the first figure being referred to, Fig.1 is simply mentioned as the illustration of visual representation learning but it lacks further explanation/discussion.**
>
> **A4:** In the revised introduction section, we refer to Fig. 1 first when we introduce instance discrimination methods and cluster discrimination methods. In the revised preliminary section, we add detailed explanations of Fig. 1 (a) and Fig. 1 (b). In Section 3.2, we also refer to Fig. 1 (c) when we introduce our method.
>
> **Q5: In Eq.1, I assume the pairwise similarity is cosine similarity if following CLIP, but without normalization of features, it is just an inner product. So I’m wondering if it is a mistake or my misunderstanding.**
>
> **A5:** We employed the normalization to all embedding features, and the pair-wise similarity is the cosine similarity. In the revised version, we added the keyword of normalization to features around Eq. (1).

---

> > ### Author Response · Authors · 2023-11-23
> > **Response to Reviewer g3TE**
> >
> > **Q6: In Eq.1, the index in the cumulative sum starts from 0 to k while that in Eq.2 is from 1 to k.**
> >
> > **A6:** In the revised version, all indices in the cumulative sum start from 1 to k. In Eq. (1), $e_j'$ contains one positive text representation for $i$ and $(k-1)$ negative text representations sourced from different instances.
> >
> > **Q7: The exponential function is denoted as exp and e at the same time in Eq.3.**
> >
> > **A7:** In the revised version, we use exp() to denote the exponential function in all equations to avoid confusion with our feature embedding symbol.
> >
> > **Q8: The ablation studies are a bit unclear to me. For example, when investigating the effects of sample ratio, the best linear probe performance is obtained when the sample ratio is set to 0.1, and the best result is 75.2. However, the linear probe performance of the proposed model shown in Table 5 is 84.6.**
> >
> > **A8:** The performance gap between Tab.5 and Tab. 6 is due to different backbones and epochs used for training.
> > In Tab. 1 - 5, the model backbone used is ViT-L/14. The epoch number is 32 (default epoch number).
> > To accelerate the hyper-parameter search, we use ViT-B/32 as the backbone and 5 epochs.
> > In the revised version, we also add the backbone information to the captions of Tab. 4, 5, and 6 to avoid confusion. Besides, we also add the epoch number into the caption of Tab. 6.

---

> > > ### Comment · Reviewer_g3TE · 2023-11-23
> > >
> > > Thank you for the clarification and the additional materials provided. Most of my concerns have been addressed but I still found it hard to understand the underlying reason why the proposed multi-label clustering idea is effective. The Figure 2 and the new Figure 5 tells a different story. To be concrete, the examples shown in Figure 2 imply that the model works because web images usually involve more than one object while multiple labels assigned to the same images can help avoid false negative supervision signals. On the other hand, Figure 5 demonstrates that the multi-labels assigned to the same images are mostly depending on the relevance between classes, rather than what is shown in the images (e.g. the class of armour is not involved in the second example in Figure 5 but it is still one of the k-nearest clusters to the target image, and the k-nearest clusters of the third example don't include a container/box even though this is explicitly shown in the target image). The experiments on ImageNet as well as the authors' discussions seem to support the latter reason but this is not consistent with the current story in the paper.

---

> > > > ### Author Response · Authors · 2023-11-23
> > > > **Discussion with Reviewer g3TE**
> > > >
> > > > First of all, great thanks for your quick reply.
> > > >
> > > > The performance increase of the proposed multi-label cluster discrimination method comes from the complementary supervision signals. It is a united story covering all cases.
> > > > For an image with multiple objects, our method can avoid false negative supervision signals. Even on the single-label dataset, ImageNet, multiple objects exist in a large number of images.
> > > > For an image including only one object, our method can improve vision representation by learning advanced concepts, e.g., attributes from different views.
> > > >
> > > > We revised the appendix section regarding multi-label learning on ImageNet.
> > > >
> > > > In Fig. 5, we visualize the top 3 labels given to some selected training samples. We cluster the original ImageNet training dataset into 4K classes by using the pre-trained CLIP model. As depicted in Fig. 5, these multiple positive labels exhibit complementary visual signals (e.g., different activities on the snow, different breeds of dogs, different locations of figs, and different objects in the room). For each of the automatically clustered 4K classes, we match the dominant class to the original 1K classes. In the single-label cluster discrimination method, only one label is used for training, and the recall of the correct label is 76.2%. By contrast, in the proposed multi-label cluster discrimination, the top 8 labels are used for training, and the recall of the correct label is 93.5%. Even though using multiple labels introduces some incorrect labels,  the proposed multi-label cluster discrimination method can learn complementary visual signals to improve visual representation learning.
> > > >
> > > > In Fig. 6, we also visualize some interesting labels given to the automatically clustered samples. As we can see, multiple positive labels sometimes show abstract correlations. For example, there is the concept of "colorful'' between "paintbrush'' and "oil painting'' in the 1st row, the concept of "shining'' between "shield'' and "armour'' in the 2nd row, and the concept of "with a handle'' between "ladle'' and "frying pan'' in the 3rd row. The concepts of "colorful'', "shining'' and "with a handle'' are class descriptions from different views. Learning from these complementary concepts can also increase vision representation learning.

---

### Official Review · Reviewer_S2Uc · 2023-11-09

**Soundness:** 3 good
**Presentation:** 4 excellent
**Contribution:** 3 good
**Rating:** 5
**Confidence:** 4

**Summary:**

In this paper, the authors propose a simple but effective method to facilitate the representation learning of the vision-language model. The method consists of two steps. In the clustering step, the authors cluster the dataset into enormous centers and utilize several closest centers as the class labels for every single image, enhancing the learning of semantic structure of training data. The discrimination step incorporates a multi-label classification loss to separate losses and promote distributed training. The experimental results are solid.

**Strengths:**

Originality: This work extends the discrimination power of CLIP model by introducing a multi-label loss to boost the semantic learning ability of the vision-language model.
Quality: The improvement achieved by the proposed method is remarkable on certain datasets, and the ablative study provides comprehensive and detailed insights into its functioning.
Clarity: This paper is reader-friendly and smooth. The experimental setting is quite reasonable.
Significance: This paper shows the benefit of using multi-label loss for clustering-based discriminative constrastive learning. This setting should be considered when developing powerful pre-trained vision-language model for downstream tasks.

**Weaknesses:**

(1) The novelty and originality of this work are limited. It seems like the method proposed in this paper incorporates several techniques introduced in the literature. It does not offer sufficient technical inspirations for the readers to follow.
(2) With respect to the limited technical novelty of this work and overall moderate improvement (I see in Table 1 and Table 2), it may not seem to be worthwhile using such huge computing resources (80 NVIDIA A100 GPUs), especially considering that visual-language pre-training field has already achieved remarkable performance.
(3) According to my understanding, as proposed method is developed upon the feature embedding from the pre-trained CLIP model and it does not involve any textual information in the proposed multiple label loss. If this is correct, this paper should make this more clear.
(4) In Equation 5, two new items are further introduced into the multi-label loss. This is regarded as one of the key contributions by this paper. However, its efficacy does not seem to be clearly verified in the ablation study. This needs to be addressed.

**Questions:**

(1) As one of the main contributions of this paper, the authors claim that the modification of optimization loss can elegantly separate the positive class labels and negative class labels, resulting in promotion of the distributed training on large-scale training data. Please explain and experimentally demonstrate how this modification can facilitate the distributed learning more clearly. For example, in Subsection Distributed Multi-label Classification of Section 3.2 MULTI-LABEL CLUSTER DISCRIMINATION, The first sentence “Eq. 6 is able to distribute the weights associated with one million class centers across all GPUs with minimal communication overhead.” Why?
(2) Some technical details are missing, e.g., in Section 4.1, the authors should explicitly point out the number of classes (k) and number of positive centers (l) they use when pre-training the model on LAION-400M dataset.
(3) In Table 3, the best results consist of both the proposed method in this paper and the FLIP (i.e., 89.1%). Notably, only the results of the proposed methodology have been highlighted.
(4) In Section 4.6, the meaning of the y-axis of the charts should be provided to improve the clarity of the paper.

**Details Of Ethics Concerns:**

No concerns. The authors have discussed the limitation at the end of the paper.

---

> ### Author Response · Authors · 2023-11-22
> **Response to the weaknesses mentioned by Reviewer S2Uc**
>
> Thanks for your valuable comments and suggestions to improve this paper.
>
> **Weakness1: novelty and originality.**
>
> **A1:**  In the instance discrimination methods (e.g., CLIP), each image-text pair represents one unique class. CLIP can provide richer forms of labels for a single image, e.g., objects, scenes, actions, and relations, at multiple levels of granularity. However, negative pairs that share similar semantics will be undesirably pushed apart in the embedding space.
>
> In the cluster discrimination methods (e.g., UNICOM), visually similar instances are pulled together by the classification step. However, UNICOM only defines a single pseudo-label for each image.
>
> To the best of our knowledge, we are the first to propose multi-label learning on the large-scale dataset (LAION-400M).
> The proposed multi-label cluster discrimination approach can not only capture the semantic structures in the data but also support the learning of multiple granularity of labels for a single image. In addition, we introduce a novel decomposition of contrastive loss, to
> avoid ambiguity during the optimization of $(s_j-s_i)$ as well as decrease the communication cost during distributed training.
>
> **Weakness2: moderate improvement.**
>
> **A2:** In this paper, we mainly compared our method with the latest state-of-the-art methods (e.g., UNICOM [1] and FLIP [2]). In the task of linear probe, our method outperforms UNICOM by 1.3% (Tab. 1). In the task of zero-shot classification, our method surpasses FLIP by 1.5% (Tab. 2). Vision representation learning is one of the most competitive research topics, and our improvement compared to this year's publications is indeed significant.
>
> [1] An, Xiang, et al. Unicom: Universal and Compact Representation Learning for Image Retrieval. ICLR 2023.
>
> [2] Li, Yanghao, et al. Scaling language-image pre-training via masking. CVPR 2023.
>
> **Weakness3:  The proposed method is developed upon the feature embedding from the pre-trained CLIP model and it does not involve any textual information in the proposed multiple label loss.**
>
> **A3:** In this paper, we focus on improving visual representation by introducing multi-label cluster discrimination. Our method does not involve text learning. In Fig. 1, we compare our method with CLIP and UNICOM. CLIP consists of the image encoder and text encoder, while UNICOM and the proposed method only contain the image encoder. In Section 4.1, we mention how to train an additional text encoder following Locked-image Tuning (LiT)[3], which teaches a text model to read out good representations from a locked image model for zero-shot classification and image-text retrieval tasks.
>
> "To assess the performance of zero-shot classification and zero-shot image-text retrieval tasks, we employ contrastive learning to train a text encoder from scratch for 32 epochs with a frozen visual encoder following LiT (Zhai et al., 2022b). The structure of the text encoder is also identical to CLIP. "
>
> [3]Zhai, Xiaohua, et al. Lit: Zero-shot transfer with locked-image text tuning. CVPR 2022.
>
> **Weakness4: In Equation 5, two new items are further introduced into the multi-label loss. This is regarded as one of the key contributions by this paper. However, its efficacy does not seem to be clearly verified in the ablation study.**
>
> **A4:**
> In the revised version, we add the experiments to verify the effectiveness of the proposed MLCD.
>
> |CASE | DATA | Finetune | Linear Probe | Zero Shot|
> | ------   | ------  | ------        | ------               | ------        |
> |MLC    | LAION-400M     | 80.9 | 76.9 | 63.9 |
> |MLCD | LAION-400M     | 81.2 | 78.1 | 64.5 |
>
> Tab. 7 (a), we compare the performance of the vanilla MLC and the proposed MLCD on the ImageNet validation dataset. Pre-training is executed on the LAION-400M dataset by 32 epochs. The model backbone is ViT-B/32.
> Both MLC and MLCD employ the negative class center sampling with a ratio of $0.1$. MLCD outperforms MLC in all three settings: fine-tuning, linear classification, and zero-shot, confirming the effectiveness of the two additional optimization targets. In Appendix. A.1, we compare their gradient calculation and time cost on the classification layer. The proposed contrastive loss decomposition can significantly decrease the communication cost, facilitating distributed training on large-scale training data.

---

> ### Author Response · Authors · 2023-11-23
> **Response to the questions arised by Reviewer S2Uc**
>
> **Q1: Please explain and experimentally demonstrate how this modification can facilitate distributed learning more clearly. “Eq. 6 is able to distribute the weights associated with one million class centers across all GPUs with minimal communication overhead.” Why?**
>
> **A1:**
> In the revised version (Appendix A.1), we add a detailed explanation regarding the gradient calculation and we also include a time cost comparison on the classification layer.
>
> As we can see, the proposed method enables an elegant separation of positive and negative gradient calculation, which can decrease the communication frequency of calling the allreduce operation.
> To compare the time cost on the classification layer, we train UNICOM, MLC, and MLCD on the LAION-400M dataset with one million classes.
> We use ViT-B/32 as the backbone, and the final embedding dimension is 512.
> The batch size is set as $32,800$ and the A100 GPU number is 80 in total distributed across 10 computation nodes.
> The time cost of the forward and backward steps regarding the backbone is $409$ms.
> For MLC and MLCD, the positive class number is set as 8.
> For the single-label classification method, UNICOM, the time cost on the classification layer is $75$ms.
> For MLC and MLCD, the time cost on the classification layer is $138$ms and $82$ms. The proposed multi-label cluster discrimination method slightly increases the time cost by 9.3% on the classification layer compared to the single-label cluster discrimination method, UNICOM. Compared to the widely used multi-label classification method, the proposed method obviously decreases the time cost by 45.6% on the classification layer. Therefore, the proposed decomposition of contrastive loss ensures efficient parallel computation across multiple GPUs from different computation nodes with minimal communication overhead.
>
> **Q2: Some technical details are missing, e.g., in Section 4.1. the number of classes (k) and number of positive centers (l).**
>
> **A2:** In the revised version, we add these technical details in Section 4.1.
> "In the following experiments, unless otherwise specified, the model used is ViT-L/14, the number of classes ($k$) is one million, and the number of positive labels ($l$) assigned to each image is $8$."
>
> **Q3: Table 3, all the best results need to be highlighted.**
>
> **A3:**   We fixed this in the revision, and we also checked other tables.
>
> **Q4: the meaning of the y-axis of the charts should be provided to improve the clarity**
>
> **A4:** In Fig. 3, we visualize per-dataset differences in the performance. The Y-axis shows the performance difference. Most of the evaluation metrics are accuracy as shown in Tab. 10 (Appendix). Green bars indicate our model outperforms the baselines, while the orange bars depict our model is surpassed by the baselines. Our model outperforms UNICOM on 23 datasets on the task of linear probe and surpasses FLIP on 15 datasets on the task of zero-shot classification. In the revised version, we added a detailed explanation in the caption of Fig. 3.

---

> > ### Comment · Reviewer_S2Uc · 2023-12-04
> >
> > Thank you for the reply and the additional materials provided. I have read the comments of other reviewers and author feedback. Basically,  most of my concerns have been addressed. But I decided to maintain the rating because of the novelty consideration.

---

### Meta-Review · Area_Chair_U62m · 2023-12-08

**Metareview:**

Based on the submission, reviews, and author feedback, the main points that have been raised are summarised as follows.

Strengths:

1. The improvement achieved is remarkable on certain datasets; the ablation study is comprehensive.
2. The paper is easy to follow; the experimental setting is reasonable;
3. The proposed algorithm is intuitive and effective; the analysis makes sense and is potentially beneficial.
4. The designed loss function elegantly separates the loss from positive and negative classes.

Issues:

1. The novelty and originality of this work are limited. It does not offer sufficient technical inspirations.
2. The efficacy of the two new items does not seem to be clearly verified.
3. Whether the top-k closest clusters to an image can really reveal what objects/attributes are involved is not clear.
4. Shall provide detailed explanation for the mathematical symbols and the distribution training process.

The authors provided feedback to address the above raised issues. All the ratings are leaning towards not accepting this work. After reading this work, I share the similar opinion. This is a solid piece of work with excellent implementation and the state-of-the-art performance. Meanwhile, the technical novelty and contribution are not sufficient for this conference. The authors could maximally incorporate the answers in the author feedback into the submission, which will well strengthen this work.

Regards, AC

**Justification For Why Not Higher Score:**

The authors provided feedback to address the above raised issues. All the ratings are leaning towards not accepting this work. After reading this work, I share the similar opinion. This is a solid piece of work with excellent implementation and the state-of-the-art performance. Meanwhile, the technical novelty and contribution are not sufficient for this conference.

**Justification For Why Not Lower Score:**

N/A

---

### Decision · Program_Chairs · 2024-01-16

Reject